# Comparing Carbon Origami from Polyaramid and Cellulose Sheets

**DOI:** 10.3390/mi13040503

**Published:** 2022-03-24

**Authors:** Monsur Islam, Peter G. Weidler, Dario Mager, Jan G. Korvink, Rodrigo Martinez-Duarte

**Affiliations:** 1Institute of Microstructure Technology, Karlsruhe Institute of Technology, Hermann-von-Helmholtz-Platz 1, 76344 Eggenstein-Leopoldshafen, Germany; monsur.islam@kit.edu (M.I.); dario.mager@kit.edu (D.M.); 2Institut für Funktionelle Grenzflächen, Karlsruhe Institute of Technology, Hermann-von-Helmholtz-Platz 1, 76344 Eggenstein-Leopoldshafen, Germany; peter.weidler@kit.edu; 3Multiscale Manufacturing Laboratory, Mechanical Engineering Department, Clemson University, Clemson, SC 29634, USA

**Keywords:** cellulose, Nomex, polyaramid, carbon, origami, lightweight, porous

## Abstract

Carbon origami enables the fabrication of lightweight and mechanically stiff 3D complex architectures of carbonaceous materials, which have a high potential to impact a wide range of applications positively. The precursor materials and their inherent microstructure play a crucial role in determining the properties of carbon origami structures. Here, non-porous polyaramid Nomex sheets and macroporous fibril cellulose sheets are explored as the precursor sheets for studying the effect of precursor nature and microstructure on the material and structural properties of the carbon origami structures. The fabrication process involves pre-creasing precursor sheets using a laser engraving process, followed by manual-folding and carbonization. The cellulose precursor experiences a severe structural shrinkage due to its macroporous fibril morphology, compared to the mostly non-porous morphology of Nomex-derived carbon. The morphological differences further yield a higher specific surface area for cellulose-derived carbon. However, Nomex results in more crystalline carbon than cellulose, featuring a turbostratic microstructure like glassy carbon. The combined effect of morphology and glass-like features leads to a high mechanical stiffness of 1.9 ± 0.2 MPa and specific modulus of 2.4 × 10^4^ m^2^·s^−2^ for the Nomex-derived carbon Miura-ori structure, which are significantly higher than cellulose-derived carbon Miura-ori (elastic modulus = 504.7 ± 88.2 kPa; specific modulus = 1.2 × 10^4^ m^2^·s^−2^) and other carbonaceous origami structures reported in the literature. The results presented here are promising to expand the material library for carbon origami, which will help in the choice of suitable precursor and carbon materials for specific applications.

## 1. Introduction

Carbon-based cellular materials are an interesting class of lightweight material due to their unique properties, including high surface-to-volume ratio, high specific stiffness, and strength, adjustable electrical and electrochemical properties, good thermal and chemical inertness, and excellent biocompatibility. These properties have enabled cellular carbon materials in several applications, such as structural materials, heat sinks, thermal insulators, electrodes for energy storage devices, catalyst supports, and scaffolds for tissue engineering [1,2,3]. Most fabrication approaches for such materials rely on the heat treatment of a shaped precursor in an inert atmosphere to achieve carbonization. At the core of most works is the understanding of the impact of the process and precursor on the microstructure and properties of the resultant carbon structure. Widely reported techniques to shape the precursor include direct foaming, templating, molding, and traditional photolithography. However, fabricating 3D complex shapes is still very challenging using these methods. In the last decade, emerging techniques including additive manufacturing and film processing have enabled the 3D structuring of porous carbon [4,5,6,7,8,9]. Examples of additive manufacturing include using two-photon polymerization-based 3D direct laser writing [8,10,11,12] to shape photosensitive materials and robocasting to extrude designed shapes out of biopolymers [13,14]. Regarding film processing, our group has reported the use of origami-inspired manufacturing for the facile and economical fabrication of 3D complex shapes of cellular carbon-based materials [15,16,17,18,19]. Origami is an ancient Japanese technique of paper folding, where a flat sheet of paper is folded along prescribed crease lines to obtain 3D complex shapes. Even though following historical traditions, origami is popular for art and decoration purposes, it has been gaining significant attention in the scientific community in recent years due to its ability to fabricate numerous 3D architectures with engineering interest [20,21]. Furthermore, origami-inspired manufacturing has advantages over the additive manufacturing routes, due to its simplicity, requiring the least technical skills, and low processing cost. Our previous publications demonstrated the fabrication of 3D carbon origami structures by folding a biopolymeric sheet into origami shapes, followed by carbonization in an inert atmosphere [15,18]. The fabricated carbon origami structures featured very low density and exhibited high mechanical stiffness at low density, which were advantageous features compared to other lightweight materials, including graphene elastomer, carbon nanotube foam, silica aerogel, and metallic microlattices [15,18]. Further work focused on demonstrating the infiltration of the biopolymeric sheets with a metal precursor to derive carbide origami structures [16] and to advance the understanding of how the processing and properties of carbon origami structures depends on the carbon precursor materials such as plant-based cellulose, bacterial cellulose, and rice paper [19].

In the present work, we expand on this work by investigating the use of polyaramid sheets for the fabrication of 3D carbon origami structures. We use Nomex (chemical name: poly (m-phenylenediamine isophthalamide)), a polyaramid typically used as a flame retardant material due to its excellent thermal, chemical, and radiation resistance properties [22,23] as well as reinforcing materials in several high-performance structural applications [22,24,25]. The carbonization of Nomex material has been studied by several researchers for the production of activated carbon materials, which have been demonstrated for applications including gas separation, catalysts, catalyst supports, and electrodes for energy storage devices [26,27,28,29]. Furthermore, recent studies have shown the feasibility of laser-induced graphitization of Nomex sheets to fabricate high-performance flexible electronics [30,31,32,33]. However, the 3D structuring of Nomex has not yet been reported.

In this work, we fabricate complex 3D shapes from Nomex-derived carbon materials using origami-inspired manufacturing. The Nomex-derived carbon material is studied for material properties and its porosity. Further, we characterize the mechanical properties of the Nomex-derived carbon origami shapes. We also perform similar experiments with cellulose paper-derived carbon origami structures to study the effect of the precursor nature and microstructure on the material and structural properties of the carbon origami structures.

## 2. Materials and Methods 

Nomex Type 410 insulation paper with a paper thickness of 130 μm was purchased from RS Components (Product number: 775–7466) and used as received. Whatman filter paper Grade 1 was used as the cellulose sheet and purchased from Sigma Aldrich (catalog number: WHA1001150). The process flow for the fabrication of carbon origami shapes is illustrated in Figure 1. We used the Miura-ori shape as the origami structure due to its shape-complexity and high interest in the engineering community [34,35,36,37]. The unit cell of a Miura-ori pattern is defined by the design parameters *h*, *l*, and *α,* as shown in the inset of Figure 1c. The Miura-ori structures fabricated here featured an *h* of 9.18 mm, an *h*:*l* ratio of 3:5, and an *α* of 41°. We pre-creased all precursor sheets using a CO_2_ laser engraving machine (ULS Versa Laser 3.50, Vienna, Austria; wavelength: 10.6 μm, lens 2.0 inch) in rastering mode (Figure 1b) at power = 2 W, speed = 12.5 cm/s, and frequency = 1000 PPI. The pre-creased precursor sheets were then manually folded into an origami structure (Figure 1d). The precursor origami shapes were then heat-treated in a tube furnace (Carbolite Gero, Neuhausen, Germany) under a constant nitrogen flow (~0.8 L/min) using the following steps: (i) raising the temperature from room temperature to 900 °C with a heating rate of 5 °C/min, (ii) dwelling at 900 °C for 1 h, and (iii) cooling down to room temperature by natural cooling. The heat treatment recipe was chosen following the recipe widely used for carbonization of other carbon precursors in carbon microelectromechanical systems (C-MEMS) technology [38,39,40,41,42,43]. The carbon materials obtained from Nomex and cellulose are denoted as NC and CC, respectively, in the rest of this article.

The microstructure of the precursor materials and carbonized samples was characterized by scanning electron microscopy (SEM, Carl Zeiss AG—SUPRA 60VP SEM). A 10 nm thick silver layer was sputtered on the precursor sheets to facilitate the imaging. Transmission electron microscopy (TEM) of the carbon materials was carried out on an FEI Titan 80–300 at 300 kV using a lacey carbon-coated copper grid with an additional 2 nm amorphous carbon coating. Raman spectrometry of the carbonized samples was performed using a Bruker Senterra equipped with DPSS laser (λ = 532 nm) at 2 mW power with a penetration depth of <1 μm. A Bruker D8 Advance diffractometer equipped with Cu-Kα_1,2_ radiation was used to characterize the crystallographic nature of the material (XRD). The porosity of the carbonized materials was characterized using nitrogen gas adsorption at 77 K with a Micromeritics ASAP 2020 instrument. The specific surface area was determined from the gas adsorption isotherms using the BET (Brunauer, Emmett and Teller) theory [44].

We calculated the structural density of the carbon Miura-ori samples using the envelope method, i.e., calculating the ratio between the mass and the volume of the carbon Miura-ori structures [9,16]. We characterized the mechanical properties of the carbon Miura-ori structures by performing compression tests in an Instron Single Column Testing System (Model 4500) by applying compressive force along Z-direction (see Figure 1e for directions). A load cell of 100 N and a compression rate of 1 mm/min were used for the compression tests.

## 3. Results and Discussion

The carbonized samples retained the original origami geometry of the precursor sheets, despite significant geometrical shrinkage. There were visual differences between the NC and the CC material. The CC material appeared dark grey and dull, whereas NC was shiny-black and featured a glass-like appearance (Figure 1e). The final dimensions of the carbon origami structures were also different despite identical precursor dimensions. The shrinkage of CC Miura-ori was considerably higher than NC Miura-ori. The shrinkage values are presented in Table 1. Both carbon Miura-ori structures featured the least shrinkage in the X-direction (see Figure 1 for the notation of the directions). For NC Miura-ori, the shrinkage values in the Y- and Z-direction were almost the same, whereas CC Miura-ori exhibited maximum shrinkage in the Y-direction. As detailed in our previous publications, the structural shrinkage of a carbon origami shape significantly depends on the dynamics of the origami shape [15,16]. Miura-ori offers the least mechanical resistance in the Y-direction, followed by Z-direction. Therefore, the highest shrinkage was expected in the Y-direction, followed by Z-direction and X-direction, which agrees with our results here. The difference in the shrinkage behavior of the investigated materials can be attributed to the microstructural features of the material.

Nomex before carbonization (Figure 2a) features fibril structures embedded in a non-porous matrix, which come from the distribution of polyaramid fibers within a polyaramid binder matrix that results from production [45]. In contrast, cellulose sheets (Figure 2d) feature randomly distributed cellulose individual fibers, with no binder in between. As illustrated in Figure 2b,e, the inherent structures of the precursors were retained during carbonization but with clear shrinkage of the fibers. The NC and CC fibers shrank ~43% and ~55%, respectively. As we speculated in our previous publication, shrinkage of the fiber network might yield pulling stresses in various directions, leading to the asymmetric structural shrinkage of the origami shape during carbonization [15]. The non-porous nature of Nomex sheet restricted the shrinkage phenomenon, which might have resulted in low pulling stress during carbonization. In contrast, the macroporous fibril arrangement of cellulose fibers led to higher shrinkage, resulting in higher pulling stress during the carbonization of the origami shapes and higher structural shrinkage. Furthermore, due to the free nature of the fibers, the fibers could slide onto each other, which might have aided the shrinkage phenomenon.

The NC sheets featured several macropores, particularly at the interfaces of the fibers and the surrounding embedding matrix, as shown in Figure 2b. These pores might have been created due to uneven shrinkage between the fibers and the matrix during carbonization. TEM investigation revealed that the NC material featured a turbostratic microstructure, exhibiting short-range graphene sheets (Figure 2c and its insets), resembling glassy carbon microstructures [46,47]. The CCs featured numerous mesopores, as visible in the top inset of Figure 2e. Furthermore, the carbon fibers featured a hollow microstructure (bottom inset of Figure 2e). The carbon fibers exhibited an amorphous-like microstructure, as characterized by TEM imaging (Figure 2f) and its respective FFT (inset of Figure 2f).

The carbon materials were further characterized by XRD and Raman spectroscopy. The XRD pattern of the NC (Figure 3a) featured two broad peaks centering around 2θ = 24° and 2θ = 44°, which are characteristics of (002), (100) and (101) reflections of turbostratic carbon material (ICDD PDF number: 75–1621) [4,46]. The XRD diffractogram of CC also featured broad peaks of carbon. However, the peak of (002) reflection shifted to 15°, and the peak of (100)/(101) appeared at 2θ = 43°. An additional peak at 2θ = 30° was observed in the diffractogram of CC, indexed to (004) carbon plane. Such differences in the XRD diffractogram of the CC can be attributed to the extended z-stack distance (L_c_) of graphitic planes of CC (L_c_ of CC = 11.93 Å versus L_c_ of NC = 6.79 Å), which may have further contributed to the higher surface area of the CC material. The Raman spectra of the carbon materials also featured two distinct peaks at 1350 cm^−1^ and 1585 cm^−1^, which correspond to the D- and G-band, respectively. The D-band represents the disorder in the carbon matrix, whereas the G-band arises from the sp^2^ carbon atoms and corresponds to the graphitic nature of the carbon material [48,49]. The intensity ratio of the D- and G-bands (I_D_/I_G_) or the ratio of the integrated areas of the D- and G-bands (A_D_/A_G_) represents the degree of graphitization of carbon material. These ratios decrease with the increased degree of graphitization and become zero for pure graphite. Here, the NC and CC featured I_D_/I_G_ of 0.95 and 0.98, respectively, whereas A_D_/A_G_ values were 2.6 and 3.1 for NC and CC, respectively. These intensity ratios suggest a relatively higher order in the NC than in the CC, which also agrees with the TEM results.

The porosity of the carbon materials produced here was characterized by nitrogen gas adsorption-desorption. The gas adsorption-desorption isotherms are presented in Figure 4a. Both the carbon materials exhibited an isotherm profile of type IV and hysteresis of H3 type, which suggests the presence of mesopores (2 nm ≤ pore diameter ≤ 50 nm) within the material. This was further confirmed by the pore size distribution, as presented in Figure 4b. Both the carbon materials featured pore sizes up to 150 nm diameter, with the majority of the pores within the mesoporous range. CC also exhibited the presence of a few micropores (pore diameter <2 nm), as shown in Figure 2b. These mesopores might have emerged due to the escape of the volatile byproducts during carbonization and localized disruption of the fibers from the surrounding matrix. Nevertheless, the gas adsorption of NC was significantly lower than CC, as evidenced by the isotherms. The higher gas adsorption of CC suggests higher pore volume and surface area for CC. The BET surface and pore volume of the carbon materials are presented in Table 2.

We characterized the mechanical properties of carbon Miura-ori structures obtained from both the precursors using compression tests. The carbon Miura-ori structures behaved differently from each other under the compressive load, as evidenced by Figure 5a,b, which present the stress-strain curve of the carbon Miura-ori samples obtained from Nomex sheet and cellulose sheet, respectively. For NC Miura-ori, the compressive stress first increased linearly with strain up to a strain of 0.04 without any mechanical failure. After that, several sudden drops and immediate steep increases were observed due to the gradual breakage of the structure under compressive load. Despite breakage, the stress value continued to increase until a strain of 0.11, which we defined as the compressive strength of NC Miura-ori. Afterward, even though repeated sharp increases and decreases in stress were observed, the peaks were considerably lower. These peaks indicated further breakage of the carbon Miura-ori samples. These peaks also signified the brittle failure of the NC origami under compressive load. Such brittle failure of NC Miura-ori can be attributed to the low porous morphology and glassy carbon-like features of NC. Once the brittle failure was initiated within the sheet, it propagated rapidly through the entire structure, causing catastrophic failure of the entire structure. In contrast, CC Miura-ori exhibited a smoother response to the compressive load. The initial part of the stress-strain curve showed a gradual increase in stress up to 0.25 strain, attributed to the levelling of the top surface of the carbon Miura-ori against the load cell surface [50,51]. After this gradual increase, a stiff and linear increase till 0.4 strain was observed, resulting in the compressive strength at this point. Further increase in strain yielded failure of the structure, resulting in a decrease in the stress. The stress again started increasing from 0.7 strain due to the densification effect. Such response to compressive load is typical for cellular materials with stretch-dominated behavior [52,53]. Under the compressive load, the load was distributed among the fibril components, which led to a smoother transition. Even though carbon is a brittle material, the macroporous fibril arrangement of CC allowed the carbon fibers to rearrange themselves to some extent prior to the brittle failure of the fibers. Furthermore, even though brittle failure of individual fibers occurred, the fiber entanglements kept the structure intact till the point of maximum strength, where the majority of the fibers reached their limits, and the entanglements started collapsing. The collapse of the entanglements yielded the postyield failure region.

Figure 5c shows the comparison of the mechanical properties and structural density of NC and CC Miura-ori structures. The CC origami structures featured a lower structural density (0.03–0.05 g/cm^3^ for CC versus 0.07–0.09 g/cm^3^ for NC) than the NC structures, which was mainly due to their differences in morphology and porosity. The elastic modulus (E) of the carbon Miura-ori structures was significantly higher for Nomex than cellulose, as shown in Figure 2c. The average value of E for NC Miura-ori was 1886.8 ± 244.2 kPa (1.9 ± 0.2 MPa), whereas CC origami exhibited an average E of 504.7 ± 88.2 kPa. It is well-known that elastic modulus scales directly with structural density [52,53]. Therefore, the higher elastic modulus of NC Miura-ori was attributed to the higher structural density of the NC Miura-ori structures. To normalize the effect of structural density, we also determined the specific modulus of the carbon Miura-ori samples by calculating the slope of the linear fit between E and density. NC Miura-ori featured a sp. modulus of 2.4 × 10^4^ m^2^·s^−2^, which was double the sp. modulus of CC Miura-ori (Figure 5d). The higher sp. modulus of NC suggests better load transfer capability within the sheets for NC, which can be attributed to the embedded fibril arrangement of NC, unlike binder-free CC fibers. The compressive strength of the carbon Miura-ori samples showed a reverse trend to the elastic modulus. The compressive strength of CC Miura-ori was significantly higher than NC Miura-ori (Figure 2c). The average strength of NC Miura-ori was 18.7 ± 8.9 kPa, compared to 61.6 ± 8.8 kPa for CC Miura-ori. As mentioned earlier, due to its non-porous and glassy nature, catastrophic brittle failure occurred for NC Miura-ori. In contrast, the fiber entanglements of CC did not allow the rapid propagation of brittle failures of individual fibers, leading to a higher compressive strength for the CC Miura-ori.

The elastic modulus (1.9 ± 0.2 MPa) and sp. modulus (2.4 × 10^4^ m^2^·s^−2^) of NC origami are significantly higher than any other carbonaceous origami structures obtained from identical precursor geometry, as depicted in Figure 5d. For example, tungsten carbide Miura-ori structures with 41° folding angle derived from a cellulosic precursor featured a sp. modulus of 0.2 × 10^4^ m^2^·s^−2^ and elastic modulus of 265.1 ± 35.8 kPa [16], whereas the sp. modulus and elastic modulus of carbon Miura-ori from chromatography paper were 0.8 × 10^4^ m^2^·s^−2^ and 193.8 ± 32.2 kPa, respectively. Furthermore, increasing the folding angle α to 75° resulted in the highest elastic modulus of 772.0 ± 100.5 kPa for the chromatography paper-derived carbon origami structures [18], which was almost 2.5 times lower than the NC Miura-ori structures reported here. However, the sp. modulus of a 75° folding angle carbon Miura-ori (2.8 × 10^4^ m^2^·s^−2^) was higher than that of NC Miura-ori reported here. It should be noted that the NC Miura-ori reported here featured the folding angle of 41°. Increasing the folding angle of NC Miura-ori is expected to significantly enhance the elastic modulus and sp. modulus. Therefore, using Nomex sheets as the precursor material is advantageous in improving the mechanical properties of carbon origami structures. Nevertheless, comparing NC and CC origami structures is relevant for placing carbonaceous origami structures as multifunctional materials. Both Nomex and cellulose have been reported several times as the carbon precursors in several applications of derived carbon materials, including gas separation, catalysts, catalysts support, and electrodes for energy-storage devices [27,28,29,54,55,56,57]. Combined with the mechanical stiffness arising from the origami structuring, the carbon origami shapes can be useful for fabricating high-performance structural filters, electrodes, and catalyst supports. However, depending on the application, one can choose the appropriate precursor for the carbon origami shapes. For example, CC origami shapes might be a better choice than NC in electrochemical applications due to the higher surface area of CC. However, NC might be more suitable in pressure gas filters due to its higher mechanical stiffness.

## 4. Conclusions

Here, we presented a comparison of cellulose and Nomex sheets as the precursor materials for fabricating carbon origami shapes. We fabricated the carbon origami shapes by pre-creasing the precursor sheets using laser engraving, followed by manual folding into 3D origami shapes and carbonization at 900 °C in an inert atmosphere. This was the first time for 3D structuring of Nomex-derived carbon material. Nevertheless, both the precursors retained the original geometry upon carbonization, despite significant shrinkage. However, the shrinkage was more severe for cellulose-derived carbon due to its fibril macroporous morphology than the predominantly non-porous morphology of Nomex-derived carbon. The difference in morphology further yielded a higher surface area for cellulose-derived carbon compared to Nomex-derived carbon. Furthermore, cellulose resulted in amorphous carbon, whereas Nomex-derived carbon featured a turbostratic microstructure, resembling glass-like carbon. The combined effect of non-porous morphology and glass-like features led to a high mechanical stiffness of 1.9 ± 0.2 MPa and specific modulus of 2.4 × 10^4^ m^2^·s^−2^ for the Nomex-derived carbon Miura-ori structure. In comparison, the cellulose-derived carbon materials exhibited an elastic modulus of 504.7 ± 88.2 kPa and a specific modulus of 1.2 × 10^4^ m^2^·s^−2^. Contrastingly, Nomex-derived carbon Miura-ori featured a significantly lower compressive strength (18.7 ± 8.9 kPa) than cellulose-derived carbon Miura-ori (61.6 ± 8.8 kPa), which was attributed to the microstructural differences between the two carbon sheets. The results presented here are promising for the expansion of the material library for carbon origami, which will help when choosing the suitable precursor and carbon materials for specific applications.

## Figures and Tables

**Figure 1 micromachines-13-00503-f001:**
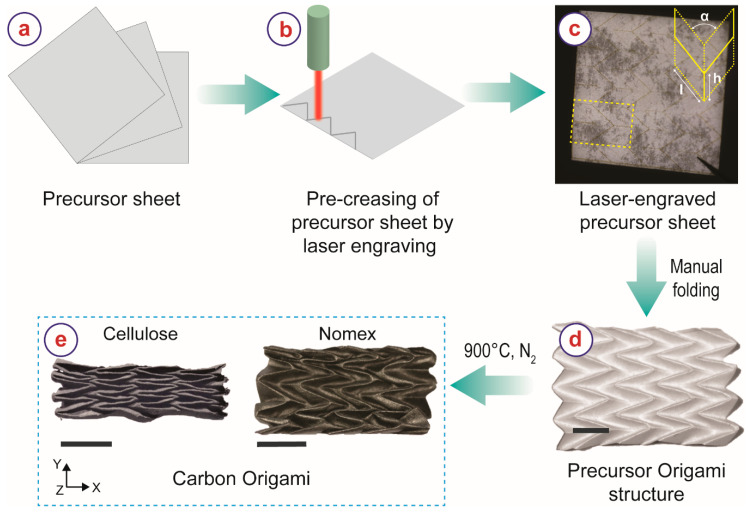
Process-flow illustrating different steps for the fabrication of carbon origami structures from cellulose and Nomex sheets. The scale bars represent 1 cm.

**Figure 2 micromachines-13-00503-f002:**
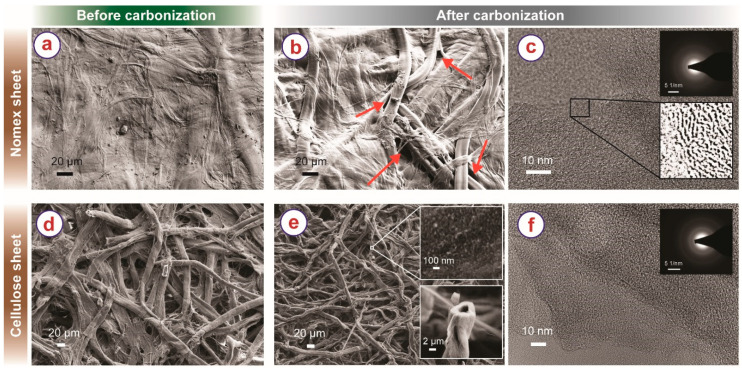
SEM of Nomex sheet (**a**) before carbonization (**b**) after carbonization. The macropores in the carbonized Nomex sheet are indicated using the arrows. (**c**) TEM of carbonized Nomex. The top inset shows the FFT of the TEM, and the bottom inset shows the magnified intensity-corrected TEM image of the selected section emphasizing the short-length graphitic layers. SEM of cellulose paper (**d**) before and (**e**) after carbonization. The top inset of (**e**) shows a higher magnification SEM image, emphasizing the porous microstructure. The bottom inset of (**e**) shows the hollow cross-section of the cellulose-derived carbon fibers. (**f**) TEM of cellulose-derived carbon fiber. Inset shows the FFT of the corresponding TEM image.

**Figure 3 micromachines-13-00503-f003:**
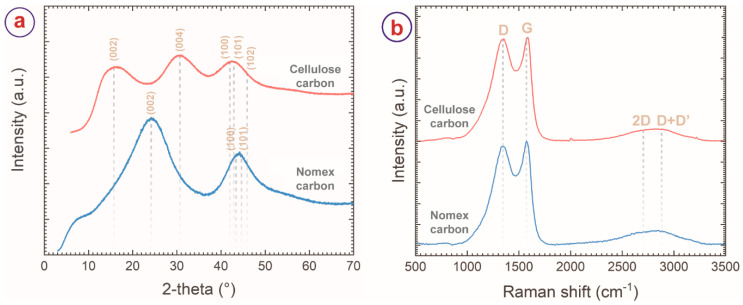
(**a**) XRD diffractograms and (**b**) Raman spectra of cellulose and Nomex-derived carbon sheets, suggesting turbostratic microstructure for both the materials, with a higher degree of crystallinity for Nomex-derived carbon material.

**Figure 4 micromachines-13-00503-f004:**
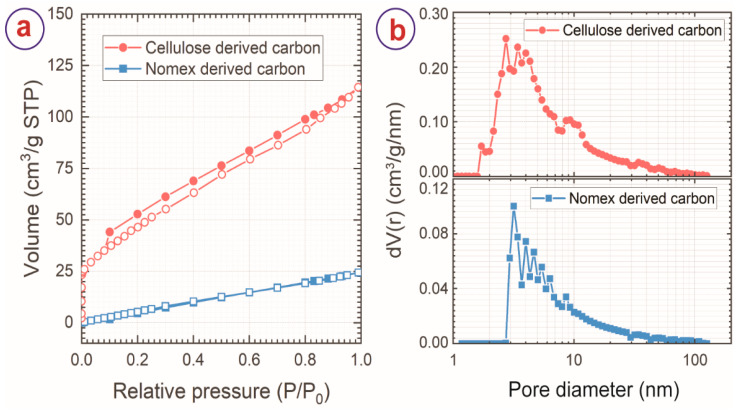
(**a**) Nitrogen adsorption-desorption isotherms and (**b**) pore size distribution of cellulose and Nomex-derived carbon sheets.

**Figure 5 micromachines-13-00503-f005:**
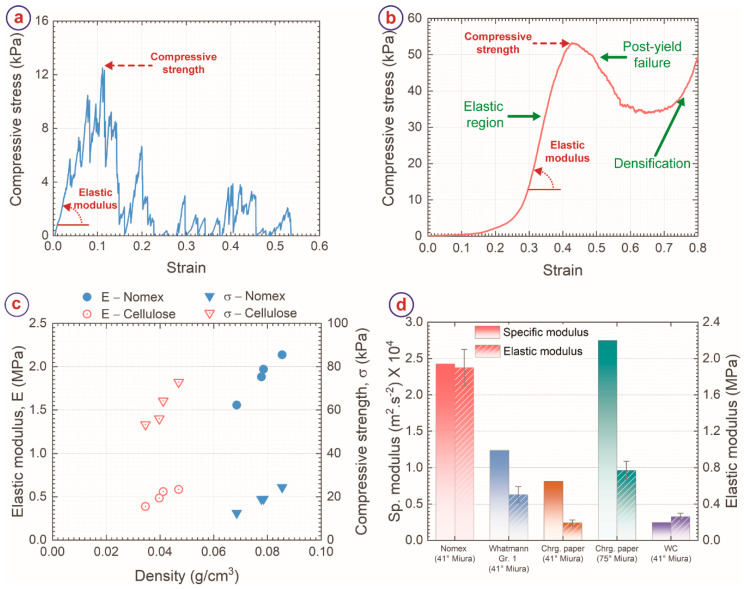
Stress-strain curve of (**a**) Nomex-derived carbon Miura-ori and (**b**) cellulose-derived carbon Miura-ori under a compressive load. The point of compression strength (σ) and definition of elastic modulus (E) were indicated in both the carbon samples. (**c**) Comparison of E and σ of the carbon Miura-ori samples against their structural density. (**d**) Comparison of E and specific modulus of different carbonaceous Miura-ori obtained from different precursor sheets. The data for chromatography (Chrg.) paper and WC were obtained from Ref. [18] and Ref. [16], respectively.

**Table 1 micromachines-13-00503-t001:** Shrinkage values of carbon Miura-ori structures derived from Nomex and cellulose sheets.

Precursor	Shrinkage (%)		
	X-Direction	Y-Direction	Z-Direction
Nomex (NC)	29.6 ± 1.6	41.7 ± 5.8	49.7 ± 2.8
Cellulose (CC)	45.4 ± 1.1	69.9 ± 4.7	51.5 ± 1.0

**Table 2 micromachines-13-00503-t002:** BET surface area and pore volume of Nomex and cellulose-derived carbon sheets.

Precursor	BET Surface Area (m^2^/g)	Pore Volume (cm^3^/g)
Nomex	44	0.04
Cellulose	180	0.16

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
