# Peer review of "Comparing Carbon Origami from Polyaramid and Cellulose Sheets"

_micromachines, 2022, doi:10.3390/mi13040503_

Round 1

Reviewer 1 Report

Comments: Authors reported Comparing Carbon Origami from polyaramid and cellulose sheets. These analyses are reasonable. However, authors should address the following comments for its acceptance.

  1. How is the synthesis method different or better than those reported earlier? What about the applicability of the resulting materials. All these points need to be explained in the introduction part.
  2. Authors need to supply the Fourier Transform Infrared and X-ray photoelectron spectroscopy to determine the functional groups and elemental composition of the Carbon Origami from polyaramid and cellulose sheets which can improve the quality of the manuscript and also for clear understanding of readers.
  3. Some of the trivial errors in the manuscript. Hence, the manuscript should be carefully checked, and necessary corrections should be done.

Author Response

REVIEWER #1:

Authors reported Comparing Carbon Origami from polyaramid and cellulose sheets. These analyses are reasonable. However, authors should address the following comments for its acceptance.

The authors thank the reviewer for taking the time to review our manuscript and giving encouraging and valuable comments, which have definitely helped us to improve the manuscript.

Comment 1. How is the synthesis method different or better than those reported earlier? What about the applicability of the resulting materials? All these points need to be explained in the introduction part.

Response: A few sentences has been added in Page 2 emphasizing the advantages of origami inspired manufacturing.

Potential applications of the carbon origami shapes reported here have been already discussed in the last paragraph of “Results and Discussion” section on Page 8.

Comment 2. Authors need to supply the Fourier Transform Infrared and X-ray photoelectron spectroscopy to determine the functional groups and elemental composition of the Carbon Origami from polyaramid and cellulose sheets which can improve the quality of the manuscript and also for clear understanding of readers.

Response: The authors thank the reviewer for this suggestion. Even though FTIR is quite accessible in our institute, it is difficult to perform XPS in a timely manner at this time of COVID-19 pandemic. Without the absence of XPS, the study of functional groups will be incomplete. Furthermore, in this study, we focus on the structural properties of the carbon origami structures, which mainly depend on the geometry and microstructure of the carbon material. In such study of structural properties, surface functional groups have minimal role. Therefore, we have not included the study of surface functional groups in this manuscript. We are nevertheless characterizing the surface functional groups for our future publication. 

Comment 3. Some of the trivial errors in the manuscript. Hence, the manuscript should be carefully checked, and necessary corrections should be done.

Response: The manuscript has been thoroughly checked and corrected the errors.

Reviewer 2 Report

The authors fabricated two sorts of carbon-based cellular materials separately by using polyaramid and cellulose sheets. Their mechanical properties were studied and discussed. This work is interesting and is helpful for the engineering community. However, the reviewer has many queries for the authors to consider carefully before recommending it for publication.

Q1. How to control the Muira-ori’s folding angle (or folding state) during the ‘Manual folding’ process?

Q2. During the heat-treated process, as stated by the authors, significant geometrical shrinkage will emerge for the origami material, which seems impossible to control precisely, so how can we guarantee that the final material meets the pre-designed requirements.  

Q3. It is not clear which direction’s mechanical properties were studied.

Q4. Stress-strain curves presented in Figures 5(a) and 5(b) are strange. For Figure 5(a), the deformation modes corresponding to the peaks or dips should be given to help better understand; while for Figure 5(b), the definition of Young’s modulus seems incorrect, and the elastic region should be the part before the strain is roughly less than 0.2.

Q5. The relationship between the macro mechanical property and the micro detection is recommended to be discussed.

Q6. This article seems to be a qualitative analysis. It is suggested that quantitative analysis can be carried out from a theoretical point of view. In this case, we will have a deeper understanding of some mechanisms.

Q7. Coordinate should be predefined before we use X-direction etc.

Author Response

The authors fabricated two sorts of carbon-based cellular materials separately by using polyaramid and cellulose sheets. Their mechanical properties were studied and discussed. This work is interesting and is helpful for the engineering community. However, the reviewer has many queries for the authors to consider carefully before recommending it for publication.

The authors thank the reviewer for taking the time to review our manuscript and giving encouraging and valuable comments, which have definitely helped us to improve the manuscript.

Q1. How to control the Muira-ori’s folding angle (or folding state) during the ‘Manual folding’ process?

Response: After manual folding, we leave the Miura-ori structures for a few hours so that they can come to a stress-free equilibrium state. At this stable state, the angles of the folded structure depend on the unit cell parameters h, l, and α, as shown in Figure 1c. These characteristics were already reported in your previous publication (Islam et al. 2016, Carbon).

Q2. During the heat-treated process, as stated by the authors, significant geometrical shrinkage will emerge for the origami material, which seems impossible to control precisely, so how can we guarantee that the final material meets the pre-designed requirements.

Response: The geometrical shrinkage depends on the dimension of the original precursor geometry and the precursor microstructure. For each precursor used here, we have tested at least 10 specimens. The standard deviations in the final dimensions are within 6%, as can be estimated from the shrinkage data reported in Table 1. Such small variation indicates the repeatability of the process.   

 Q3. It is not clear which direction’s mechanical properties were studied.

 Response: The compressive load was applied along the Z-axis of the Miura-ori (see Figure 1e for directions) during the compression tests.

Q4. Stress-strain curves presented in Figures 5(a) and 5(b) are strange. For Figure 5(a), the deformation modes corresponding to the peaks or dips should be given to help better understand; while for Figure 5(b), the definition of Young’s modulus seems incorrect, and the elastic region should be the part before the strain is roughly less than 0.2.

Response: Figure 5a presents an example of a stress-strain curve of Nomex-derived carbon (NC) Miura-ori. NC exhibited a highly brittle nature under compressive load, similar to glass-like carbon. Due to such brittle nature, during the compression test, NC Miura-ori started breaking into small pieces. The peaks in the stress-strain curve represent the point of catastrophic brittle failure. We have already discussed this in the manuscript.

We disagree with the reviewer regarding Figure 5b. The gradual increase in stress up to 0.25 strain is attributed to leveling of the top surface of Miura-ori with the load cell surface. Such behavior was also reported in other publications. We have added this information in the manuscript with proper citations (Ref 50, 51) on Page 7.

Q5. The relationship between the macro mechanical property and the micro detection is recommended to be discussed.

 Response: The relationship between the macrostructural properties and microstructural properties is indeed a topic of our interest. However, the current manuscript does not focus on investigating such a relationship, rather it focuses on the comparison of the polyaramid and cellulose precursor for the fabrication of carbon origami. We have tried to qualitatively discuss the influence of microstructure of these two precursors on their respective mechanical properties on Page 7. However, we acknowledge that the reported experiments in this manuscript are not enough to understand or explain the relationship. More extensive study is needed to fully elucidate this relationship.

Q6. This article seems to be a qualitative analysis. It is suggested that quantitative analysis can be carried out from a theoretical point of view. In this case, we will have a deeper understanding of some mechanisms.

 Response: The authors thank the reviewer for the suggestion. Our ongoing work includes the numerical investigation of the mechanical properties of carbon origami structures, which we intend to publish in our future publication. Therefore, we are not including the quantitative analysis in this manuscript. We appreciate the understanding of the reviewer.

Q7. Coordinate should be predefined before we use X-direction etc.

Response: The coordinates are already defined in Figure 1e.

Round 2

Reviewer 1 Report

The authors have addressed the comments. The manuscript has been improved.

Reviewer 2 Report

I recommend it for publication since most of my concern has been well addressed.